# Lithiophilic-lithiophobic gradient interfacial layer for a highly stable lithium metal anode

Huimin Zhang[1,2], Xiaobin Liao[3,4], Yuepeng Guan[5], Yu Xiang[1], Meng Li[1], Wenfeng Zhang[1], Xiayu Zhu[1], Hai Ming[1], Lin Lu[1], Jingyi Qiu[1], Yaqin Huang[5], Gaoping Cao[1], Yusheng Yang[1], Liqiang Mai [3], Yan Zhao [4] & Hao Zhang[1]

The long-standing issue of lithium dendrite growth during repeated deposition or dissolution processes hinders the practical use of lithium-metal anodes for high-energy density batteries. Here, we demonstrate a promising lithiophilic–lithiophobic gradient interfacial layer strategy in which the bottom lithiophilic zinc oxide/carbon nanotube sublayer tightly anchors the whole layer onto the lithium foil, facilitating the formation of a stable solid electrolyte interphase, and prevents the formation of an intermediate mossy lithium corrosion layer. Together with the top lithiophobic carbon nanotube sublayer, this gradient interfacial layer can effectively suppress dendrite growth and ensure ultralong-term stable lithium stripping/plating. This strategy is further demonstrated to provide substantially improved cycle performance in copper current collector, 10 cm$^2$ pouch cell and lithium–sulfur batteries, which, coupled with a simple fabrication process and wide applicability in various materials for lithium-metal protection, makes the lithiophilic–lithiophobic gradient interfacial layer a favored strategy for next-generation lithium-metal batteries.

[1] Research Institute of Chemical Defense, 100191 Beijing, China. [2] Beijing Institute of Technology, 100081 Beijing, China. [3] State Key Laboratory of Advanced Technology for Materials Synthesis and Processing, Wuhan University of Technology, 430070 Wuhan, China. [4] State Key Laboratory of Silicate Materials for Architectures, Wuhan University of Technology, 430070 Wuhan, China. [5] State Key Laboratory of Chemical Resource Engineering, The Key Laboratory of Beijing City on Preparation and Processing of Novel Polymer Materials, Beijing University of Chemical Technology, 100029 Beijing, China. These authors contributed equally: Huimin Zhang, Xiaobin Liao. Correspondence and requests for materials should be addressed to L.M. (email: mlq518@whut.edu.cn) or to Y.Z. (email: yan2000@whut.edu.cn) or to H.Z. (email: dr.h.zhang@hotmail.com)

ithium (Li) metal is the preeminent anode choice for Li batteries due to its ultrahigh theoretical capacity of 3861 mAh g$^{-1}$ and the most negative potential among all the electrode materials[1–3]. Despite the successful use in primary batteries[1–4], Li-metal electrodes have a poor cyclability and encounter severe safety concerns caused by the formation of Li dendrites[5–7] and a low Coulombic efficiency[8–10] in secondary batteries. To realize rechargeable Li-metal anodes, growing research efforts since the 1960s have been devoted to understanding the process of Li deposition fundamentally and to suppressing the constant growth of dendrites, which can bring about thermal runaway and explosion hazards from short circuits, as well as inferior cyclability from an unstable solid electrolyte interphase (SEI)[9–16].

In the last ten years, the soaring interest in Li–sulfur and Li–air batteries has intensified these efforts of suppressing dendrite growth, and the associated research can be classified into five categories[17–19]: (i) replacing Li metal with a LiX alloy (X = Al, Si, C, etc.) to alleviate the concerns of dendrite growth, (ii) designing high-modulus solid electrolytes (including inorganic, polymer, and hybrid) to suppress dendrite penetration[20–22], (iii) optimizing electrolyte components (especially additives for SEI stabilization) or developing stable modified interfaces to reinforce SEI formation and prevent dendrite propagation[13,23,24], (iv) manipulating the nanoarchitectures of the Li-metal anode and minimizing electrode dimension variation by stable hosts, skeleton structures, or metal current collectors[15,25–29], and (v) constructing a robust and electrochemically stable upper interfacial layer for Li-metal anodes[10,30–36]. All five strategies are effective for improving SEI stability and suppressing dendritic Li growth to some extent. However, the first three are limited to solving the infinite volume change caused by the intrinsic problems of "hostless" Li deposition/dissolution, which is particularly urgent for practical applications of Li anodes[17,18]. In addition, the performance of solid electrolytes remains unsatisfactory because such electrolytes often exhibit inferior ionic conductivity and a large interfacial resistance[20–22]. Furthermore, the liquid electrolyte additives used for SEI stabilization are gradually drained during battery cycling[11,13,37,38]. The approaches that use prestored Li three-dimensional (3D) structured anodes show attractive properties because of the reduced volume variation and lower effective current density associated with increasing the active Li surface. However, based on our previous work, the long-term SEI stabilization in a high-surface-area 3D nanostructured Li anode is questionable, and dendrites still tend to form after the inner space is filled by Li deposits[28]. In addition, the challenges associated with large-scale fabrication of such nanostructured anodes with a tailored thickness by a simple process remain formidable in the practical application of Li-metal batteries.

Relative to the other four methods, the interfacial layer strategy is a promising option to address dendrite growth and enable large-scale fabrication and application. Several carbon morphologies (nanosphere, nanotube, graphene, and fiber), ceramics (fluoride, nitride, phosphate), polymers, and their composite interfacial layers have been proven successful in regulating the deposition of Li and finally preventing dendrite growth[10,30–36]. However, the design guidelines and mechanism interpretation are still empirical, and the prerequisite for an ideal Li anode interfacial layer is still elusive. The reported interfacial layers in the literature are very different in both chemical and physical properties, including ionic and electronic conductivity, porosity, lithophilicity, and mechanical strength. Therefore, to achieve high-energy Li-metal batteries, it is imperative to elucidate physicochemical properties regarding Li dendrite suppression, further optimize the interfacial layer, and design an effective strategy to fabricate an ideal structure while ensuring long-term cyclability for the Li-metal anode.

Herein, we develop a lithophilic–lithophobic gradient strategy by dripping carbon nanotubes (CNT) with various ZnO loadings layer by layer onto Li foil (termed GZCNT) based on our experimental discovery that there are essential properties, of which lithophobicity, mechanical robustness, and Li ion conductivity are the most critical, for a Li metal interfacial layer (Fig. 1). The bottom layer of lithophilic zinc oxide/CNT (ZnO/CNT) tightly anchors the entire layer to the Li foil, facilitates the formation of a uniform SEI, and eliminates the mossy Li corrosion layer between them. In addition, the top layer of lithophobic CNT with a porous morphology facilitates Li diffusion and is robust enough to avoid the penetration of Li dendrites. Thus, the gradient interfacial layer can suppress dendrite growth effectively and ensure ultralong-term stable Li stripping/plating even at a high current density with a high Li capacity, facilitating the integration of various advantages from Li protection strategies using the reported carbon materials. This strategy is further demonstrated to be successful in copper (Cu) current collector, 10 cm$^2$ pouch cell and Li–sulfur (Li–S) batteries. More remarkably, this lithophilic–lithophobic gradient strategy is applicable to carbon materials such as CNT and can accommodate other materials for Li anode modification such as incorporation of electrospun fibers.

## Results

**Properties of Li foils with various interfacial layers.** Figure 2 illustrates the synthesis and morphology of Li-metal anodes with various upper interfacial layers. CNT, ZnO/CNT, graphene, carbon black, or carbon fiber were used as the starting materials. We first dry these materials in vacuum at 120 °C for 2 h, move them into an Ar-filled glove box, disperse them in 1,3-dioxolane (DOL) with a concentration of 0.2 wt% by stirring for 12 h, drip the suspensions onto the Li foils with 100 μL cm$^{-2}$ by a pipette and then dry using a hot plate at 80 °C for 1 h, corresponding to a mass loading of 0.2 mg cm$^{-2}$ for all the layers. Cross-sectional scanning electron microscope (SEM) images (Supplementary Figs. 3a, 5a, and 6a) show that the thickness of graphene, ZnO/CNT, and CNT layers are 35, 20, and 25 μm, respectively. The thickness of electrospun fiber was ~10 μm; this layer was prepared by a conventional electrospinning method as reported previously[25]. Transmission electron microscope (TEM) imaging, X-ray photoelectron spectroscopy (XPS), SEM imaging, and corresponding C, O, and Zn elemental energy dispersive X-ray (EDX) mapping imaging of ZnO/CNT (Supplementary Fig. 1c, d, e)

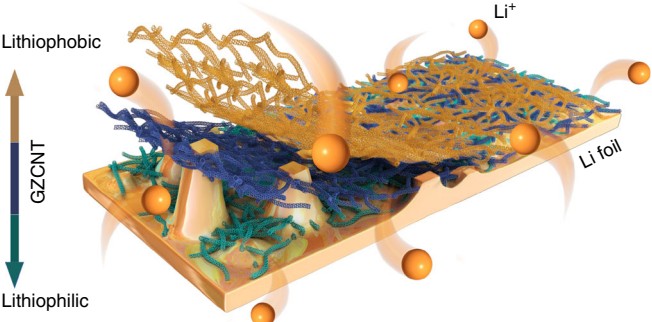

**Fig. 1** Schematic diagram for Li deposition of Li foils coated with a GZCNT interfacial layer. The bottom lithophilic part of the GZCNT layer anchored onto the Li foil and effectively ensured even Li plating by regulating deposition, while the top lithophobic part maintained a porous morphology to facilitate Li diffusion and hinder dendrite formation, resulting in the formation of a dendrite-free Li-metal anode

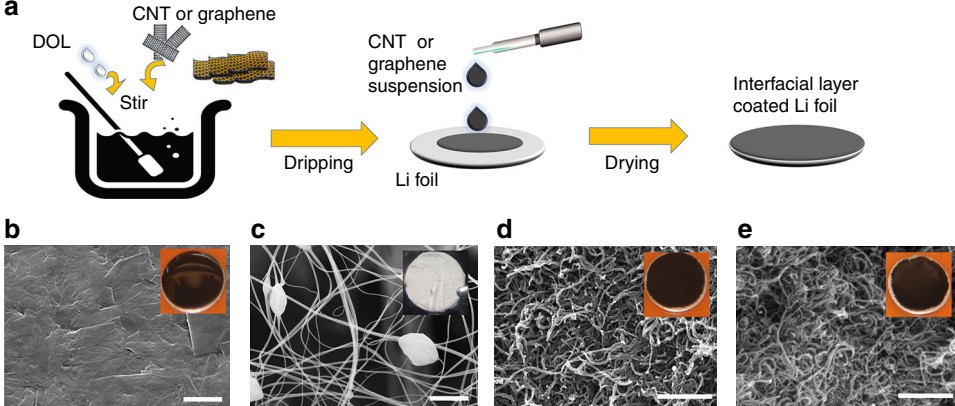

**Fig. 2** Synthesis and characterization of Li metal anodes with various upper interfacial layers. **a** Schematic of interfacial layer fabrication in a glove box. Graphene, ZnO/CNT, or CNT were mixed with DOL, and the suspension was dripped onto the Li foil at 100 μL cm⁻² and dried at 80 °C for 1 h. **b–e** Top-view SEM images of **b** graphene, **c** electrospun fiber, **d** ZnO/CNT, and **e** CNT interfacial layer. The insets show digital photographs of the corresponding Li foils with interfacial layers. Scale bar, 2 μm for **b** and **c**, and 500 nm for **d** and **e**

reveal that ZnO particles are finely dispersed on the surface of single CNT. The mass percent of ZnO in ZnO/CNT is 29.8 wt% (Supplementary Fig. 1f), which is comparable to the content of ~25 wt% for the Li matrix modified by ZnO/porous carbon that was recently found to improve Li storage performance[29]. The graphene layer (Fig. 2b) presents a much more compact microstructure than the other three layers shown in Fig. 1. The pore size distributions of graphene and CNT layers derived from combined $N_2$ sorption and mercury intrusion porosimetry, shown in Supplementary Fig. 1a, confirm that the graphene layer features more developed micropores (<2 nm) and fewer mesopores than the CNT layer. Raman spectra (Supplementary Fig. 1b) demonstrates that the graphene and CNT have similar $I_G/I_D$ ratios (9.6 and 9.1) and exhibits similarly favorable crystallization properties. Note that the heat treatment at 1500 °C for 8 h can effectively enhance the crystallization and reduce the oxygen (from 3.75 to 2.77%, Supplementary Fig. 1c) and nitrogen (from 2.69 to 0%) functional groups.

We conducted electrochemical testing of a blank Li foil cell and the cells assembled by Li foils with various interfacial layers using 2032-type two-electrode symmetric coin cells with an ether-based electrolyte (0.6 M lithium bis(trifluoromethanesulfonyl)imide (LiTFSI) and 0.4 M $LiNO_3$ in 1:1 w/w DOL/dimethoxyethane (DME)) (Fig. 3). The electrochemical impedances of all pristine cells (Fig. 3a, b) reveal that the blank cell has a large $R_{SEI}$ (242 Ω) relative to cells with Li foils coated by graphene (123 Ω), electrospun fiber (98 Ω), ZnO/CNT (48 Ω), and CNT (71 Ω). The much smaller $R_{SEI}$ is attributed to the much better wettability between the electrode and electrolyte created by the interfacial layer. At a current density of 1 mA cm⁻² (Fig. 3c), the Li|Li symmetrical cell exhibited a larger voltage hysteresis during Li deposition/dissolution (>50 mV), which increased sharply from the 102nd cycle, indicating the failure of the cell due to exhaustion of the electrolyte caused by repeated SEI fracture/formation. In contrast, the Li foils coated with electrospun fiber and graphene showed an improved cycle life of 145 and 180 cycles, respectively. The ZnO/CNT-coated device obtained a much better performance of 275 cycles, and the CNT-coated device obtained the best result of 520 cycles. Generally, it is accepted that DOL can facilitate the formation of a relatively flexible oligomer SEI, thus improving the long-term cycling stability of Li-metal anodes[39], while the bare Li electrode and the Li coated by electrospun fiber still exhibit remarkable necking behaviors (the overpotential decreases at first, and then increases) during cycling (Fig. 3c), which is characteristic of stabilization of

the SEI in the early stage and subsequent SEI accumulation from the continuous growth of dendritic Li[40]. The difference in overpotential and long-term cyclability became more prominent at a higher current density of 5 mA cm⁻² (Fig. 3d). The voltage hysteresis for the pristine Li foil cell was ~500 mV during stripping/plating and immediately exhibited voltage fluctuations. In contrast to the cycle instability of the bare Li cell, the cell with ZnO/CNT-coated Li foils cycled stably for nearly 40 h, while the CNT-coated one remained healthy up to the 210th cycle (84 h) and showed a lower overpotential (~200 mV). We present the electrochemical performance of cells assembled by Li foils with carbon fiber and carbon black interfacial layers in Supplementary Fig. 2. Notably, the cycle stability enhancement of carbon fiber interfacial layer was very limited, and the carbon black layer even deteriorated the cycle stability, although the $R_{SEI}$ of Li foils coated with carbon black (76 Ω) and carbon fiber (81 Ω) was remarkably lower than that of the blank one and was comparable to the $R_{SEI}$ of CNT-coated Li foils (Supplementary Fig. 2b, c).

**Failure and prerequisite of interfacial layers**. The SEM images (Fig. 4 and Supplementary Figs. 3–8) reveal the top-view morphology of the interfacial-layer-coated Li foils after extended stripping/plating at 1 mA cm⁻² with a cycle capacity of 1 mAh cm⁻². Notably, only the CNT electrode displayed a well-retained morphology (Fig. 4d) over 1000 h, which is attributed to its highly porous and mechanical robust properties and lithiophobic nature (Supplementary Fig. 6). The CNT interfacial layer exhibited a high Young's modulus of 8 GPa, i.e., 40 times higher than that of the electrospun fiber layer (Supplementary Fig. 6e, f). CNT also presented a low binding energy of ~1.19 eV with a Li atom (Supplementary Fig. 6g, h), which is remarkably lower than the binding energy of Li with copper (2.57 eV), graphene (3.64 eV), and nitrogen functional groups (~4 eV) on carbon, demonstrating the lithiophobic nature of the CNT after high-temperature treatment.

For the graphene-coated Li foil, Li could directly deposit on the upper surface of graphene layer and form whisker-shaped elongated Li deposits (Fig. 4a and Supplementary Fig. 4b–d), and the graphene layer was ultimately coated by Li dendrites (inset of Fig. 4a). Relative to the CNT layer, the graphene layer presented similar chemical properties (Raman spectra in Supplementary Fig. 1b) but far fewer mesopores (Supplementary Fig. 1a), which may hinder Li diffusion through interfacial layer and resulted in the formation of Li deposits on the upper surface instead of the lower one. For the Li foil coated with an

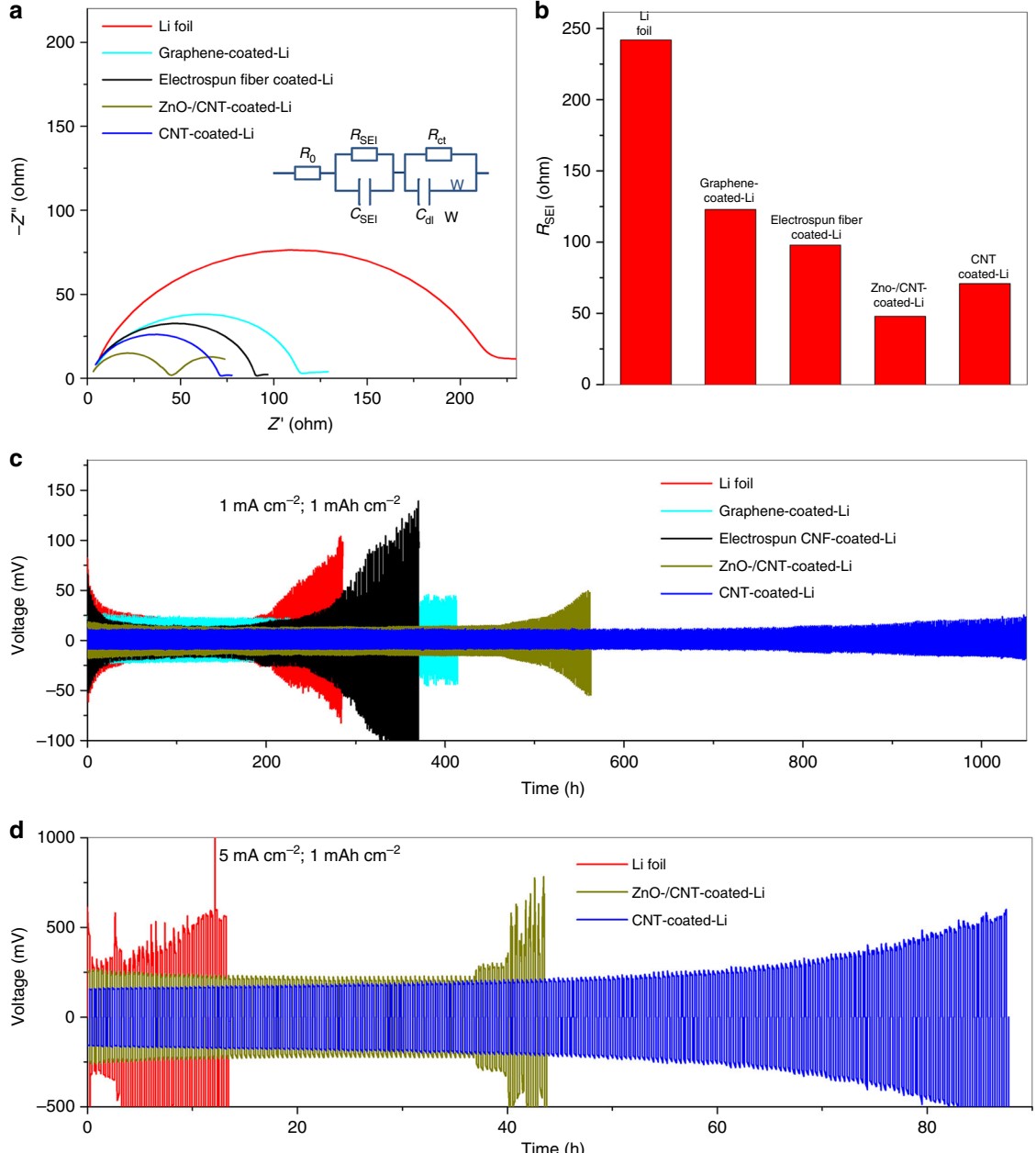

**Fig. 3** Electrochemical performance of a blank cell and the cells assembled by Li foils with various interfacial layers. **a** Electrochemical impedance spectra of various cells (inset in Fig. 2a: equivalent circuit for fitting the impedance results). Frequency: 0.1 Hz–1 MHz, perturbation amplitude 5 mV, measured at 0% state of charge (SOC). **b** Comparison of $R_{SEI}$ fitting results of all cells. **c**, **d** Comparison of the cyclability of a symmetric cell assembled by blank Li foils (red) and Li foils with interfacial layers of graphene (light blue), electrospun fiber (black), ZnO/CNT (dark yellow), and CNT (blue), respectively, at current densities of **c** 1 and **d** 5 mA cm$^{-2}$ with a deposition/dissolution capacity of both 1 mAh cm$^{-2}$ for **c** 520 and **d** 220 cycles

electrospun fiber layer, although its porosity was as developed as that of the CNT layer, most of the electrospun fiber layer was buried underneath the Li deposits after 180 cycles, which is attributed to its flexible and soft properties that allow easy bending and coating by newly grown Li deposits (Fig. 4b and Supplementary Fig. 4b–d, in which the boundary between Li deposits and remaining electrospun fiber is prominent). For the ZnO/CNT-layer-coated Li foil, Li could be easily deposited on the surface of ZnO-coated CNT, which is lithiophilic (Supplementary Fig. 9), and the interspaces in the layer were slowly occupied by Li deposits as cycling continued (Supplementary Fig. 5b, c). Mossy Li tended to form on the stuffed ZnO/CNT interfacial layer, leading to the failure of the cell (Fig. 4c and Supplementary

Fig. 5d). Because both carbon fiber and carbon black have unfavorable film-forming properties, the interfacial layers formed by them were discontinuous (insets of Supplementary Figs. 7a and 8a); thus, they failed to protect the Li foils from uneven Li deposition (the carbon black layer even worsened this condition). In brief, all the interfacial layers in Fig. 4, including carbon materials and electrospun fiber, were unable to stabilize the SEI over the long term, consequently resulting in an ever-increasing interfacial resistance due to repeated crack/formation in the SEI.

Through analysis of the shortcomings presented above, we identify an experimental discovery of the guidelines for ideal interfacial layer design. The interfacial layer should be lithiophobic and robust to avoid the penetration of Li dendrites, and it

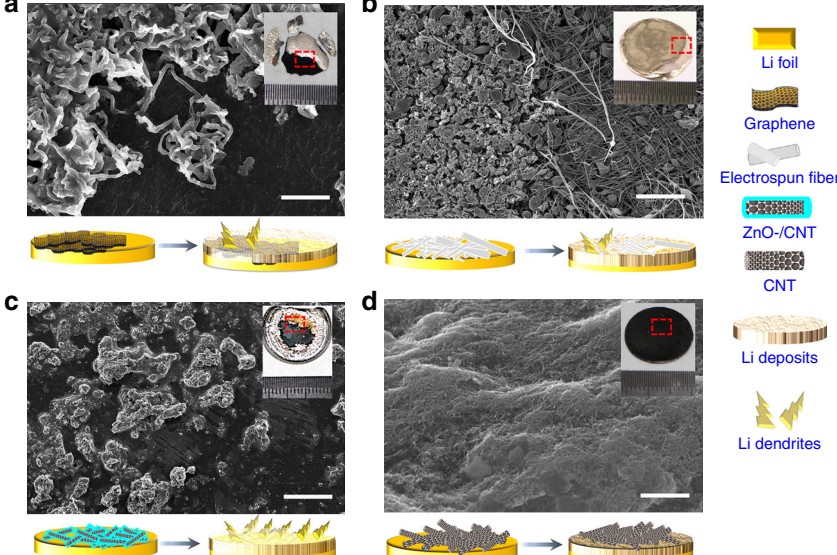

**Fig. 4** SEM images of top view morphology of the interfacial layer-coated Li foils after extended stripping and plating. Li foil symmetric cells coated with **a** graphene, **b** electrospun fiber, **c** ZnO/CNT, and **d** CNT, cycled at 1 mA cm$^{-2}$ with a capacity of 1 mAh cm$^{-2}$ after (**a**) 210, **b** 180, **c** 280, and **d** 520 cycles. The insets show digital photographs of the corresponding Li foils with interfacial layers after testing. The schematics under each figure show the Li deposition/dendrite growth process. Scale bar, 20 μm for **a**, 5 μm for **b**, 200 μm for **c**, and 2 μm for **d**

should be porous to ensure the fast diffusion of Li ions. We present the prerequisites for an ideal Li foil interfacial layer in Supplementary Fig. 10.

**Preparation and characterization of the GZCNT interfacial layer.** Even the best interfacial layer, i.e., the lithiophobic CNT-coated Li foil, still suffered from voltage divergence after 200 cycles at 5 mA cm$^{-2}$ current density, and the voltage rapidly exceeded 500 mV (Fig. 3d), indicating continuous Li dendrite growth, substantial pulverized "dead" Li formation, repeated SEI crack/formation and internal microshorting. The cross-sectional SEM images of the corresponding CNT-coated Li foil after 220 cycles indicate that an irregular but distinct gap appeared between the CNT layer and Li foil (Supplementary Fig. 6d). Although no whisker-shaped dendrites were visible on either side of the CNT layer, the gap was fully occupied by a Li deposit layer of up to 100 μm thick considering that the original CNT layer was 20 μm in thickness (inset of Supplementary Fig. 6a). This thick degradation layer reflects the uneven Li deposition, which resulted in exhaustion of the electrolyte and failure of the cell. Thus, it is important to design a rational and subtle strategy to further eliminate the degradation of this layer and thus enhance the long-term cycling stability.

Inspired by the phenomenon that the lithiophobic CNT layer blocks dendrites and that lithiophilic ZnO/CNT ensures perfect infusion of Li into the porous layer, we present a novel strategy of creating a lithiophilic–lithiophobic gradient interfacial layer by dripping ZnO/CNT with different ZnO loadings layer by layer onto Li foil. As shown in the schematic of Supplementary Fig. 11, three suspensions with 0.2 wt% ZnO/CNT, 0.1 wt% ZnO/CNT and 0.1 wt% CNT, and 0.2 wt% CNT were dripped onto the Li foil with a mass loading of 40, 30, and 30 μL cm$^{-2}$, respectively, to obtain the GZCNT interfacial layer. Cross-sectional SEM images and the corresponding Zn elemental EDX mapping data (Fig. 5) reveal that the ZnO loading of the 20-μm-thick pristine GZCNT layer gradually increases from top to bottom. The upper, middle, and lower parts of the GZCNT layer all present a highly porous morphology, with a Zn content

of 0.39, 6.87, and 14.53 wt%, respectively (Fig. 5), demonstrating the ZnO gradient loading property. Note that it is important to present the mass and volume fractions of the active material for more comprehensive electrode performance analysis[41]. The thickness (20 μm), volume, and weight (0.2 mg cm$^{-2}$) fraction of the GZCNT layer were less than 0.85%, 0.85%, and 0.19%, respectively, and had almost no influence on the performance of the whole electrode.

**Suppression of dendrite Li metal growth during deposition.** Electrochemical performance of cells assembled by Li foils with normal CNT and GZCNT interfacial layers are compared with bare Li foils in Fig. 6. The symmetrical cell of bare Li failed at the 102nd cycle at 1 mA cm$^{-2}$ (Fig. 6a). In contrast, the GZCNT-coated cell obtained the best stability of 520 cycles, while the CNT-coated cell tended to exhibit a slight voltage divergence starting at the 400th cycle. The difference in cyclability and voltage hysteresis became more distinct at higher current densities of 5 and 10 mA cm$^{-2}$ (Fig. 6b, c). The CNT-coated Li foil survived for 60 h at 5 mA cm$^{-2}$ but immediately exhibited voltage fluctuations leading to failure at 10 mA cm$^{-2}$. By contrast, the cell with GZCNT-coated Li foils could stably cycle for more than 120 and 100 h at 5 and 10 mA cm$^{-2}$, respectively. In addition, typical Li stripping/plating profiles at 1 and 10 mA cm$^{-2}$ (Supplementary Fig. 12a, b) indicate that the voltage hysteresis of the GZCNT-coated Li cell is much smaller than that of the CNT-coated Li cell. To obtain high-energy-density batteries (400 Wh kg$^{-1}$ or higher), safely accepting and releasing 5 mAh cm$^{-2}$ capacity for Li anode is indispensable[17]. Here, we demonstrate a galvanostatic cycling stability test with a capacity of 5 mAh cm$^{-2}$ at 1 mA g$^{-2}$ (Fig. 6d). In striking contrast, the bare Li-metal anode exhibited sharp voltage fluctuations from the first few cycles, which is attributed to an internal short circuit or electrolyte depletion from repeated Li deposition and dissolution. The voltage hysteresis of the CNT-coated Li foil cell was ~20 mV and slowly increased to ~40 mV (Supplementary Fig. 12c). The GZCNT-coated cell maintained a much lower plating/stripping overpotential of 10 mV for more than 50 cycles. Such superior cycle stability is a promising

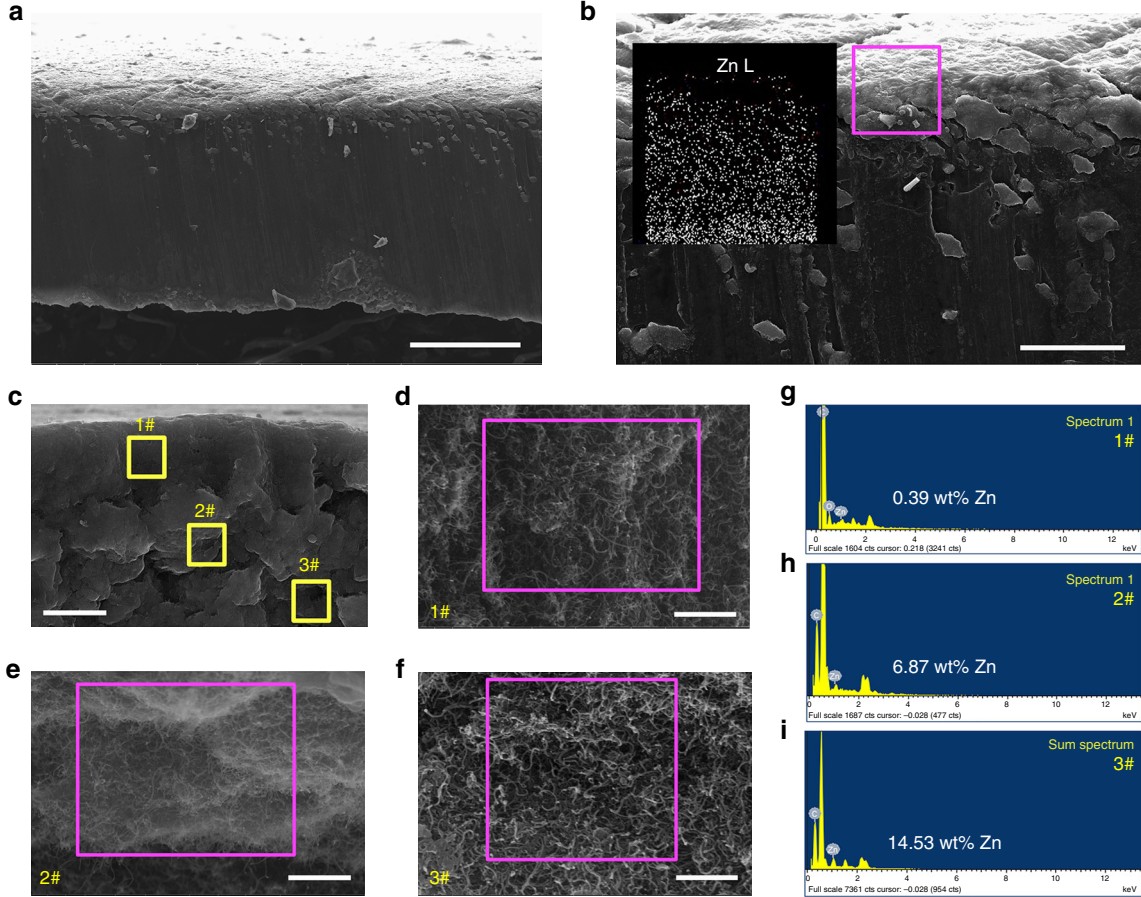

**Fig. 5** Characterization of the gradient ZnO-coated CNT (GZCNT). **a–f** Cross-sectional SEM images of **a, b** pristine GZCNT-coated Li foil and **c–f** the GZCNT interfacial layer. Inset in **b** is the corresponding Zn (Zn–L) elemental EDX mapping image of GZCNT interfacial layer highlighted by pink. Morphology of #1 upper (**d**), #2 middle (**e**), and #3 under (**f**) parts of layer highlighted by yellow in **c**. **g–i** Corresponding elemental EDX spectrum of GZCNT interfacial layer, indicating that the Zn content of the **d** upper, **e** middle, and **f** lower parts of the layer are **g** 0.39, **h** 6.87, and **i** 14.53 wt%, respectively. Scale bar, 100 μm for **a**, 20 μm for **b**, 10 μm for **c**, and 1 μm for **d–f**

indicator of effective regulation of Li deposition/dissolution without dendrite formation, promoting its promising application future in high-energy-density rechargeable batteries. The highly decreased polarization and impressive cyclability can be further verified by the electrochemical impedance analysis spectrum (EIS) using symmetric cells after 500 cycles (Fig. 6e). Although both the GZCNT- and CNT-coated Li cell Li foils showed similar small $R_{SEI}$ values of 74 and 71 Ω, respectively, their $R_{SEI}$ after 500 cycles changed in opposite ways. The $R_{SEI}$ of the CNT-coated cell increased to 158 Ω, while that of the GZCNT-coated cell decreased to 30 Ω (Fig. 6f) due to the formation of a stable and ion-conducting SEI. This result demonstrates that a GZCNT-coated Li electrode contributes greatly to favorably stable Li ion-diffusion kinetics.

The limitations of the CNT layer and advantage of the GZCNT layer are elucidated by the SEM images and schematics shown in Fig. 7. The cross-sectional SEM images of the CNT-coated Li clearly show that a >20 μm gap is generated between the CNT layer and Li foil that is filled by irregular shaped Li deposits (Fig. 7b). The highly porous morphology is maintained, and no Li dendrites are detectable either on the surface or in the CNT layer, suggesting that the porous and lithiophobic CNT layer is highly permeable to Li ions. Although no dendrites are visible, the mossy Li deposits in the gap tend to result in exhaustion of the electrolyte and failure of the cell (Fig. 7i). By contrast, the GZCNT layer eliminates this Li degradation layer by the lithiophilic ZnO/CNT layer (Fig. 7e, f). SEM images (Fig. 7g, h) confirm that after 520

cycles, Li is tightly deposited and no crevices or dendrites are detectable, while the porous upper CNT layer is not filled with deposits, which is ideal for long-term stability of Li anodes.

To further elucidate the Li deposition process, we show SEM images of the cross-sectional morphology of pristine GZCNT-coated Li foil and the same foil after stripping and plating back 1 mAh cm$^{-2}$ and after 500 cycles at 1 mA cm$^{-2}$ with various magnifications in Supplementary Fig. 13. As illustrated by the schematic of Fig. 7j, the top lithiophobic part of the GZCNT layer retained a porous morphology to facilitate Li diffusion and hinder dendrite formation (Supplementary Fig. 13e, f, h), while the bottom lithiophilic layer was anchored to the Li foil and effectively ensured even Li plating by regulating deposition, resulting in the fact that neither dendrites from the electrode nor a corrosion layer between the CNT and Li foil were formed.

It is demonstrated that the gradient strategy also excels at a copper current collector that delivers an outstanding cyclability, low voltage hysteresis, and a remarkable Coulombic efficiency of approximately 99.5% for 100 cycles, even with a higher cycling capacity of 3 mA h cm$^{-2}$ at 2 mA cm$^{-2}$ (Supplementary Fig. 16). The strategy may not be limited to various carbon materials such as CNT and may be extended to other materials for Li metal modification. Here, the strategy was typically applied to the gradient ZnO/electrospun fiber (GZF) interfacial layer (Supplementary Figs. 18–20) and proved equally effective in regulating Li deposition. As shown in Supplementary Fig. 17, the bottom lithiophilic layer anchored tightly onto the Li foil, effectively

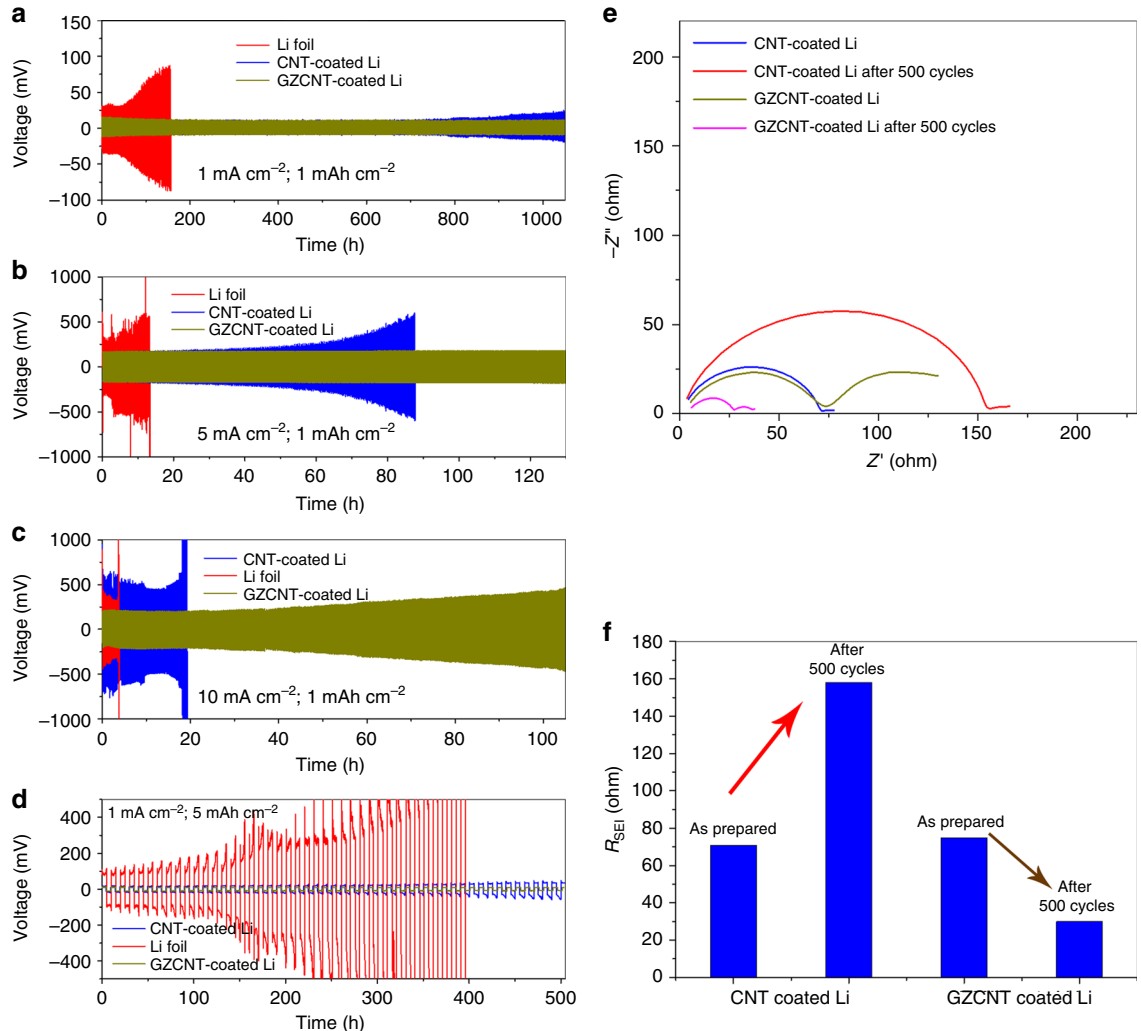

**Fig. 6** Electrochemical performance of cells assembled by Li foils with a GZCNT interfacial layer. **a–d** Comparison of the cyclability of a symmetric cell assembled by blank Li foils (red), and Li foils with interfacial layers of CNT (blue), and GZCNT (gold) at current densities of **a** 1, **b** 5, and **c** 10 mA cm$^{-2}$ with a stripping/plating capacity of 1 mAh cm$^{-2}$ for **a** 520, **b** 325, and **c** 520 cycles, and **d** at 1 mA cm$^{-2}$ with 5 mAh cm$^{-2}$ capacity for 50 cycles. **e** Electrochemical impedance spectra of pristine and post-500-cycle (at 1 mA cm$^{-2}$ with a charging/discharging capacity of 1 mAh cm$^{-2}$) cells assembled by Li foils with CNT and GZCNT interfacial layers. Frequency: 0.1 Hz–1 MHz, perturbation amplitude: 5 mV, measured at 0% SOC. **f** Summary of $R_{SEI}$ fitting results of pristine and post-500-cycle cells

facilitating stable SEI formation and regulating Li deposition. Together with the top lithiophobic CNT, this GZF interfacial layer can effectively suppress dendrite growth even under a higher cycling density of 5 mA cm$^{-2}$ and ensure ultralong-term stable Li deposition/dissolution.

To probe the potential practical applications of the GZCNT interfacial layer for large-area Li-metal anodes, we assembled symmetric Li foil pouch cells with a surface area of 10 cm$^2$. Relative to the coin cell, more than 10 times the current was applied to a pouch cell, readily resulting in prominent and unevenly distributed Li dendrites (Supplementary Fig. 14a). The bare Li pouch cell suffered from voltage fluctuations starting at the 55th cycle, while the GZCNT-coated Li pouch cell exhibited stable cycling for more than 200 cycles, with no dendrite visible on the electrode (Supplementary Fig. 14b). The stable long-term cyclability and significantly reduced polarization can be further confirmed by the EIS (Fig. 8d, e). The GZCNT cell obtained a lower $R_{SEI}$ of 2.46 Ω, which decreased slightly to 2.21 Ω after 200 cycles. By contrast, the $R_{SEI}$ of blank Li coin cell was 4.51 Ω and increased to a high resistance of 1537 Ω after 200 cycles due to the exhaustion of the electrolyte from repeated SEI breakdown/

reformation (reflected by the distinct degradation layer shown in Fig. 8c).

To further demonstrate the dendrite suppression effect of the GZCNT layer, we built an in situ optical observation symmetric cell. We tested the blank and GZCNT-coated Li foils before and after Li stripping/plating at 10 mA cm$^{-2}$ with 1 mAh cm$^{-2}$ capacity for 200 cycles; digital photographs are shown in Supplementary Fig. 15. The GZCNT layer clearly protected the Li anode from dendrite formation. For the blank Li foils, the pristine smooth Li foils generated large dendrites after 200 cycles[42]. Relative to the blank Li, the GZCNT-coated Li enabled a remarkable visual result: half of the foil (with bare Li) was coated by large dendrites, and the half with the GZCNT-coated area remained black and smooth with no visible dendrite formation, confirming the effect of the GZCNT layer on dendrite suppression.

To demonstrate the lithiophilic–lithiophobic gradient strategy for high-energy batteries, we assembled Li–S batteries for comparison. A high areal sulfur loading up to 2.5 mg cm$^{-2}$ was used to ensure a high areal capacity of ~3 mAh cm$^{-2}$. Relative to the Li–S cell with blank Li foil, the GZCNT-coated Li cell showed

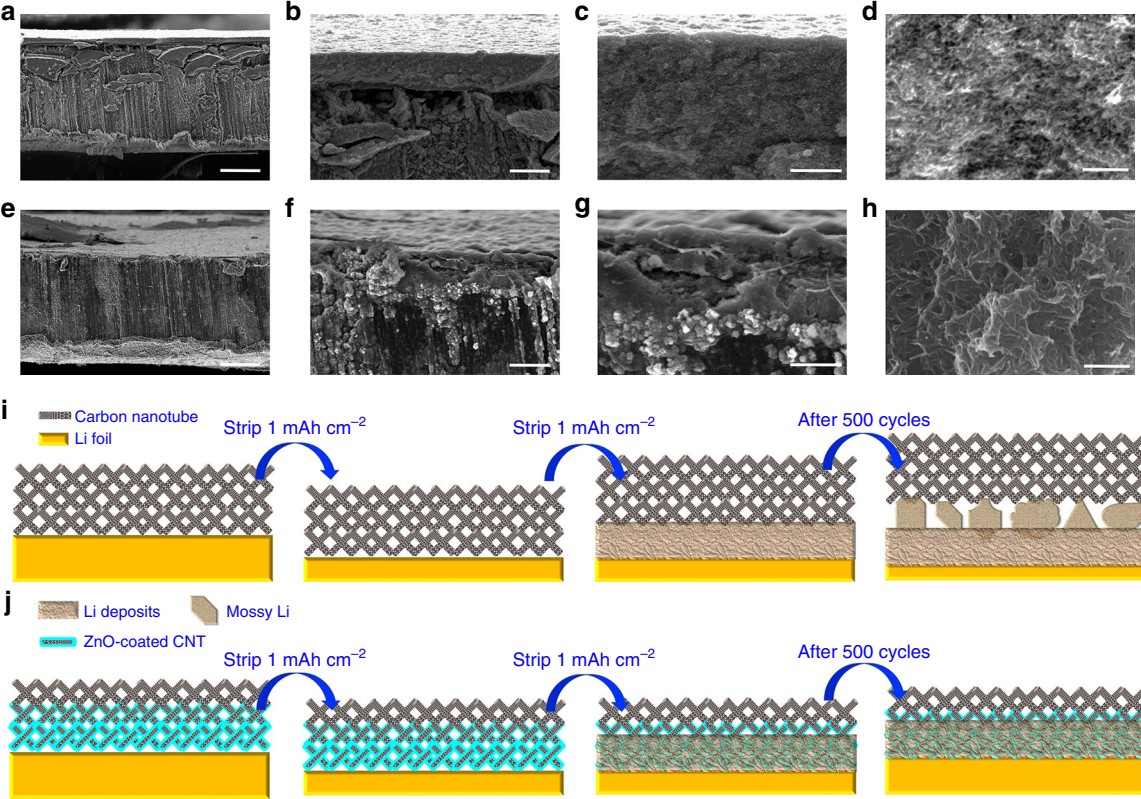

**Fig. 7** Well-confined stripping and plating behavior of the GZCNT-coated Li foil. **a–h** SEM images with various magnifications of cross-sectional view morphology of the Li cells coated with a CNT (**a–d**) or GZCNT (**e–h**) interfacial layer and cycled at a current density of 1 mA cm$^{-2}$ with a stripping/plating capacity of 1 mAh cm$^{-2}$ after 520 cycles. **i**, **j** Li stripping/plating mechanism of Li foils coated with a **i** CNT and **j** GZCNT interfacial layer where, after extended stripping/plating, a mossy Li layer forms between the CNT layer and Li foil (**i**), while the Li deposits nucleate at the surface of the underlying lithiophilic part of GZCNT, leading to a tightly anchored Li/CNT interface while the upper lithiophobic layer that avoids the deposition of Li, consequently avoiding dendrite growth from the electrode and any corrosion layer between the CNT and Li foil. Scale bar, 100 μm for **a**, **e**, 20 μm for **b**, **f**, 5 μm for **c**, **g**, and 1 μm for **d**, **h**

a comparable initial capacity but clear improvement in cycle stability at 0.2 C (Fig. 9a, b). After 200 cycles, the remained capacity for blank cell and GZCNT cell was 1.12 and 1.73 mAh cm$^{-2}$, respectively. The charge/discharge profiles are displayed in Fig. 9c to probe the reasons for this improvement. A two-stage discharge profile appears for both cells, reflecting the reduction process of sulfur to Li-polyS and Li-polyS to Li sulfides at the voltage platforms of ca. 2.31 and 2.10 V, respectively. The capacity decaying during cycling occurred over both stages. In addition, both the high and low voltage platforms decreased, and the effect was much more distinct for the blank Li–S cell, showing a largely increased polarization. To characterize the Li-ion diffusion kinetics, the initial EIS and that after 200 cycles were obtained. Relative to the resistance increase of the GZCNT-coated Li cell from 77 to 164 Ω, the resistance of the blank Li–S cell increased from 79 to 372 Ω (Fig. 9b), demonstrating a much larger resistance to Li diffusion and subsequent severe polarization. The anode surface was extensively buried under mossy Li deposits after 200 cycles in the blank Li–S cell (Fig. 9e), and the Li degradation layer was too thick and then broke away from the Li foil (inset of Fig. 9e), forming a dead Li layer, severely consuming the liquid electrolyte and increasing the resistance substantially. By contrast, no Li dendrite was visible on the GZCNT-coated Li after 200 cycles, and the porous nature of the upper lithiophobic CNT layer was maintained and could effectively sustain fast Li diffusion to the deposition sites.

## Discussion

In summary, we reveal an experimental discovery that lithiophobicity, mechanical robustness, and favorable Li ion diffusion are three essential properties for a stable Li-metal interfacial layer suppressing dendrite formation by a comparison of six widely used configurations. We have found that even the best interfacial layer, i.e., lithiophobic CNT, failed to effectively regulate Li deposits over the long term because a mossy Li layer tended to form underneath the CNT interfacial layer. Based on these insights, which deepen the understanding of the mechanism of Li metal stripping/plating on the upper interfacial layer, we have developed a lithiophilic–lithiophobic gradient strategy. The bottom lithiophilic ZnO/CNT tightly anchors the whole layer onto the Li foil and facilitates the formation of a stable and ion conductive SEI that eliminates the mossy Li corrosion layer at the interface. Together with the top lithiophobic CNT, this gradient interfacial layer can effectively suppress dendrite growth at a higher areal current density of 10 mA cm$^{-2}$ and ensure ultralong-term stable Li stripping/plating. This strategy was further demonstrated to enable significantly improved cycle performance in Cu current collector, 10 cm$^2$ pouch cell and Li-S batteries. Notably, this lithiophilic–lithiophobic gradient strategy not only is applicable to most carbon materials such as CNT but also can accommodate other materials for Li anode modification such as electrospun fibers. This capability, together with the simple fabrication process, makes the lithiophilic–lithiophobic gradient interfacial layer a promising

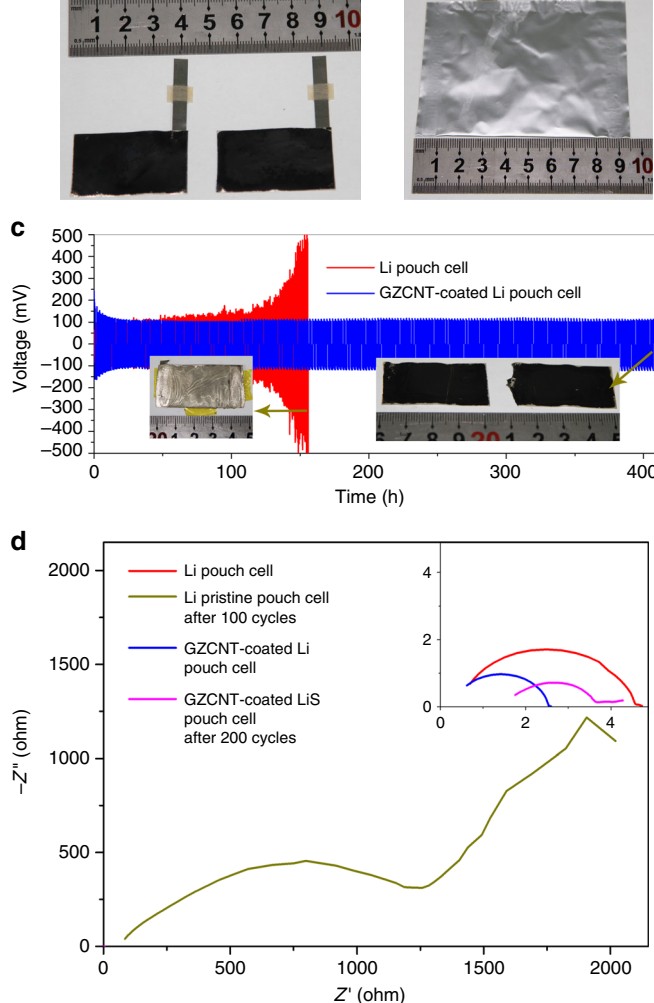

**Fig. 8** Electrochemical performance of 10 cm$^2$ pouch cells assembled by Li foils with GZCNT interfacial layers. **a** Digital photos of two 40 × 25 mm$^2$ GZCNT-coated Li foils and **b** the pouch cell assembled by them. **c** Comparison of the cycling performance of symmetric pouch cells assembled by blank Li foils (red), and Li foils with GZCNT (blue) at a charge/discharge current density of 1 mA cm$^{-2}$ (10 mA in total) with a stripping/plating capacity of 1 mAh cm$^{-2}$ (10 mAh in total) for 210 cycles. The insets are digital photographs of Li foil (left) and GZCNT-coated Li foils (right) in pouch cells after 100 and 200 cycles, respectively. **d** Electrochemical impedance spectra of pristine pouch cells assembled by bare Li foils and GZCNT-coated Li foils and after 100 and 200 cycles. Frequency: 0.1 Hz–1 MHz

strategy for next-generation Li-metal battery systems such as Li–S, Li–air, Li–polymer, and Li–intercalation oxides.

## Methods

**Preparation of Li foils coated with interfacial layers**. Graphene (Moxi Tech Corp., China), carbon black (Jinpu Corp., China), and carbon fiber (Xiangsheng Corp., China) were used as received. The CNT (Cnano Tech Corp., China) used in the experiment were all heated at 1500 °C for 2 h to remove the nitrogen functional groups before use and enhance the graphitization degree of CNT, which may facilitate fast electron transport. ZnO/CNT was synthesized by a chemical precipitation method in which Zn(OAc)$_2$·2H$_2$O (Aladdin, 99%) was chosen as the Zn source, LiOH·H$_2$O (Aladdin, 98%) as the precipitant, and ethanol as the solvent. First, two kinds of solution were deployed as follows: 40.0 mg CNT and 116.0 mg Zn(OAc)$_2$·2H$_2$O were dissolved into 25 mL ethanol to obtain solution A. Then, 46.0 mg LiOH·H$_2$O was immersed into 25 mL ethanol to get solution B. Solution B was dripped into solution A slowly through a 100 mL injector under the condition of continuous stirring, forming a black gel. Subsequently, the ZnO/CNT

nanocomposite (with a ZnO content of 29.8 wt%) was collected after filtrating, washing, and drying at 180 °C for 12 h. For the preparation of electrospun fiber, a homogeneous polyimide (PI) solution (15 wt%) was obtained by dissolving PI powder (DuPont CP-0650) in N-methyl-2-pyrrolidone while stirring in a 60 °C bath overnight. The solution was transferred into a glass syringe with a stainless-steel needle tip connected to a high voltage supply (ES30P-5 W, Gamma High Voltage Research). The distance between the needle tip and the collector (graphite paper) was set as 15 cm, and the solution pumping rate was 10 mL min$^{-1}$. The voltage applied on the needle was set to 15 kV; notably, to ensure the homogeneity of the electrospun film, a negative voltage of 1 kV was applied to the collector. The thickness of the electrospun PI matrix obtained was ~20 μm with 4 mg cm$^{-2}$. The electrospun fiber was dried and coated directly on the Li foil in a glove box.

CNT, ZnO/CNT, graphene, carbon black, or carbon fiber were the starting materials; they were first dried in vacuum at 120 °C for 2 h and then moved them into an Ar-filled glove box, dispersed them in DOL with a concentration of 0.2 wt% by stirring for 12 h, dripped the suspensions onto the Li foils with 100 μL cm$^{-2}$ by a pipette and then dried the product using a hot plate at 80 °C for 1 h, leading to an areal mass loading of 0.2 mg cm$^{-2}$ for all the layers. For the preparation of the GZCNT interfacial layer, three suspensions with 0.2 wt% ZnO/CNT, 0.1 wt% ZnO/CNT and 0.1 wt% CNT, and 0.2 wt% CNT were subsequently dripped onto the Li foil by a mass loading of 40, 30, and 30 μL cm$^{-2}$ (ref. [41]). The GZF layer was prepared by depositing ZnO on the electrospun PVDF fiber (purchased from China Nano Technology Co., Ltd.) by an RF magnetron sputtering system (PD 280) under argon with high purity for 40 min, during which the RF power was powered under 300 W with the substrate temperature at room temperature. The GZF layer (Supplementary Figs. 21–22) was obtained by depositing ZnO on the electrospun PVDF fiber by an atomic layer deposition system (Cambridge NanoTech Savannah S100). The deposition process was maintained for 500 cycles under 80 °C. The ZnO loading layer was deposited on both the top and back of the fibers at a rate of 0.1 nm per cycle.

**Structural and characterization of interfacial layers**. Raman spectroscopy (LabRAM HR800, JY Horiba) and X-ray diffractometry (XRD) (Bruker/D8 advance) were used to obtain structural information. Thermal gravimetric analysis (TGA) was performed using a simultaneous thermal analyzer (NETZSCH STA 409 PC Luxx, Germany) from 30 to 800 °C at a heating rate of 5 °C min$^{-1}$ under a N$_2$ atmosphere. High-resolution transmission electron microscopy (HRTEM) and selected-area electron diffraction (SAED) patterns were obtained using a TEM (FEI-Tecnai G2 F30). The SEM images and corresponding EDX elemental mapping were recorded on an SEM (FEI XL30 Sirion) at an acceleration voltage of 5 kV. Specially, for Li surface morphology observation after cycling, the electrodes were disassembled from the coin cell and gently rinsed using DOL in the glovebox. The cross sections of the samples were imaged at a slightly tilted angle for convenience of obtaining the morphology and EDX spectrum. The surface elemental composition was detected by Perkin-Elmer PHI-5300 ESCA XPS. The pore structure information was obtained using a Micromeritics ASAP 2020 adsorption apparatus at 77 K and at a pressure of up to 1 bar after 24 h degasification at 180 °C. The specific surface area was calculated using the Brunauer–Emmett–Teller model in a relative pressure ranging surface. Mercury intrusion porosimetry was performed using an AutoPore IV 9500, and the Young's modulus of the interfacial layers was obtained by atomic force microscopy (AFM) (Bruker Multimode 8 with a Nanoscope V controller) in an Ar-filled glove box.

**Computational details**. The calculations were conducted with density functional theory (DFT) using the CASTEP code in Materials Studio (version 2017 R2) of Accelrys, Inc. The exchange-correlation functional employed was the Perdew–Burke–Ernzerhof (PBE) generalized gradient approximation (GGA). The convergence tolerance of geometry optimization was set to (1) $1.0 \times 10^{-5}$ eV atom$^{-1}$ for energy, (2) $3.0 \times 10^{-2}$ eV Å$^{-1}$ for maximum force, and (3) $1.0 \times 10^{-3}$ Å for maximum displacement. The k-points for Brillouin zone sampling were selected by the Monkhorst–Pack method and set to $6 \times 6 \times 1$ for the CNT models. In addition, the energy cutoff was set as 950 eV for the CNT–Li model, and the self-consistent field (SCF) tolerance was set as $1.0 \times 10^{-6}$ eV atom$^{-1}$. A $4 \times 1 \times 1$ super cell with 1.5 nm vacuum was used for the CNT model. The model was assigned to interact with Li atoms on the surface, and the corresponding binding energy was defined as follows:

$$E_{binding} = E_{total} - E_{sub} - E_{Li},$$

where $E_{total}$, $E_{sub}$, and $E_{Li}$ are the energies of the CNT model bound with a Li atom, the bare CNT model, and the Li atom, respectively.

**Electrochemical measurements**. Li deposition and dissolution processes were investigated using a symmetric configuration in 2032-type coin cells. The electrolyte employed contained 0.6 M LiTFSI dissolved in 1:1 w/w DOL/DME with the addition of 0.4 M LiNO$_3$. The separator used was Celgard 2325. The control symmetric Li|Li cells were assembled by freshly scraped Li foil. Charging/discharging testing was performed on a Wuhan LAND battery testing system. The impedance analysis was conducted on a Solartron 1280 test station.

For the preparation of pouch cells, two 40 × 25 mm$^2$ GZCNT-coated Li foils with a GZCNT loading of 0.2 mg cm$^{-2}$ was assembled into one cell in a glove box,

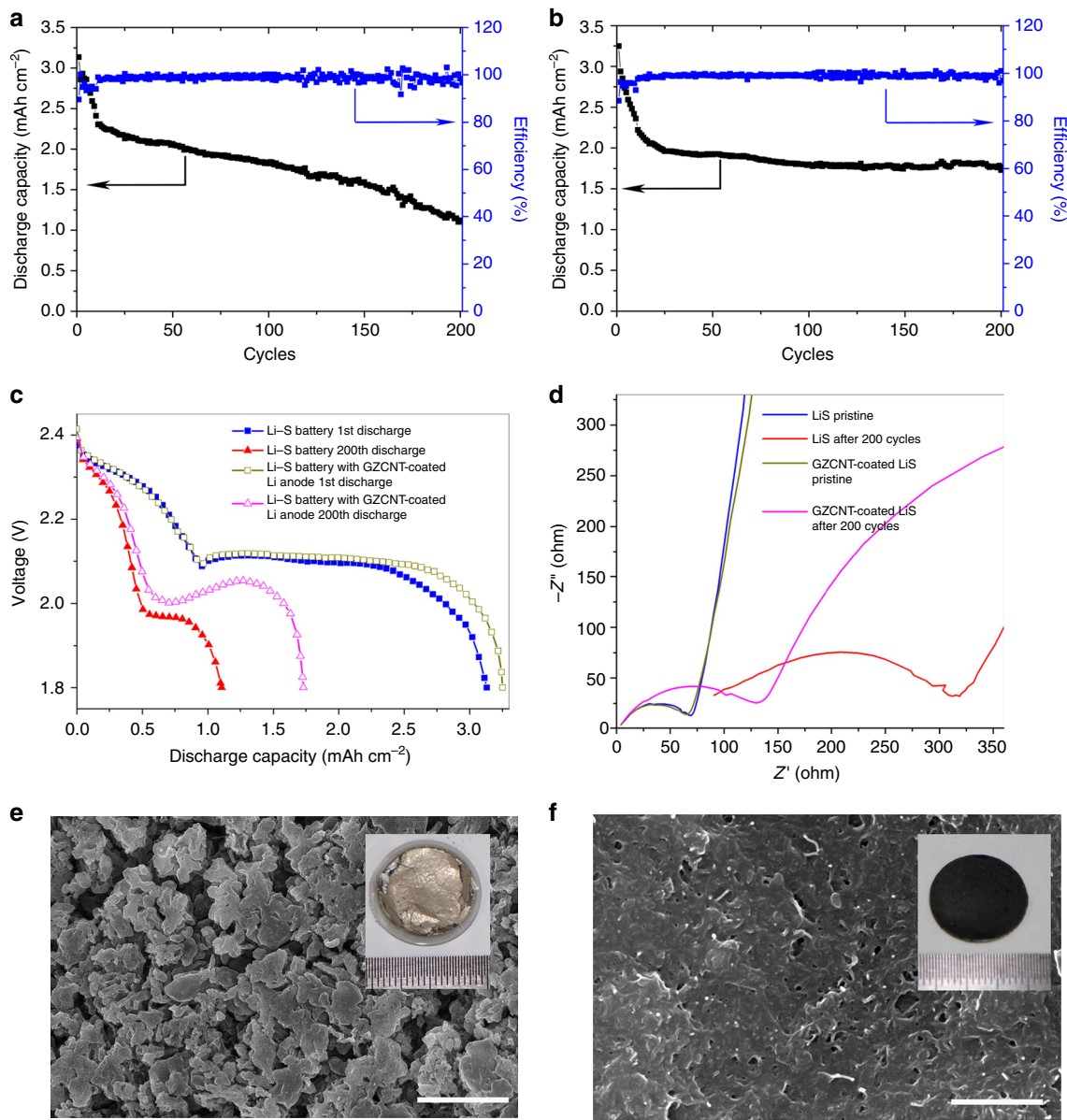

**Fig. 9** Electrochemical performance of Li–S batteries with the GZCNT-coated Li foil anode. **a–c** Charge–discharge properties at 0.2 C (0.6 mA cm⁻²) with a capacity of up to 3 mAh cm⁻². Discharge capacity and Coulombic efficiency of Li–S batteries with **a** a pristine Li foil and **b** the GZCNT-coated Li foil anode, and **c** charge and discharge curves of pristine and after-200-cycle Li–S coin cells. **d** Electrochemical impedance spectra of pristine and after-200-cycle Li–S coin cells assembled from blank and GZCNT-coated Li. Frequency: 0.1 Hz–1 MHz, perturbation amplitude: 5 mV, measured at 100% SOC. **e**, **f** Top-view SEM images of the blank (**e**) and GZCNT-coated Li foils (**f**) from Li–S coin cells after 200 cycles. The insets show the digital photos of the corresponding Li anodes without (**e**) and with (**f**) GZCNT interfacial layers after 200 cycles. Scale bar, 5 μm for **e** and 1 μm for **f**

with Celgard 2325 as a separator and 1.0 mL 0.6 M LiTFSI and 0.4 M LiNO₃ in 1:1 w/w DOL/DME as an electrolyte.

For the preparation of sulfur cathode in Li–S cells, elemental sulfur, conductive agent (acetylene black), and binder (LA132, 15 wt%) were mixed in a solution of *n*-propyl alcohol and deionized water in a weight ratio of 80:10:10, then stirred for 12 h to obtain a homogeneous mixed slurry that was coated on an aluminum current collector by an automatic coating machine and then dried at 60 °C for 24 h in a vacuum oven before use; the diameter of the electrode disks was 14 mm. The areal sulfur mass loading in each electrode was approximately 2.5 mg cm⁻².

For the preparation of the in situ observation cell, a pair of blank (20 × 5 mm²) and half-GZCNT-coated Li foils (20 × 10 mm², with approximately the entire area coated by GZCNT) were assembled into a glass vial with ~5 mL 0.6 M LiTFSI and 0.4 M LiNO₃ in 1:1 w/w DOL/DME as an electrolyte, respectively.

## Data availability

The data supporting the findings of this work are presented in this article and its supplementary information files.

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

## Acknowledgements

H.Z. acknowledges support from the National Key Research and Development Program of China (No. 2016YFB0901503). L.M. acknowledges funds from National Key Research and Development Program of China (2016YFA0202603), the National Natural Science Fund for Distinguished Young Scholars (51425204), the Program of Introducing Talents of Discipline to Universities (B17034), the National Natural Science Foundation of China (51521001), and the Fundamental Research Funds for the Central Universities (WUT: 2016III001). All the authors acknowledge assistance in AFM testing by Miss Shuangyan Lang and Prof. Rui Wen from the Institute of Chemistry, CAS.

## Author contributions

L.M. and H.Z. conceived the idea. H.M.Z. and Y.G. carried out the materials synthesis and electrochemical characterization. X.L. and Y.Z. carried out theoretical calculations. Y.G. carried out the TEM measurements. Y.Z., Y.X., M.L., W.Z., X.Z., H.M., L.L., J.Q., Y.H., G.C., and Y.Y. provided important experimental insights. H.M.Z. wrote the paper. All the authors discussed the results and contributed to writing the manuscript.

## Additional information

**Competing interests:** The authors declare no competing interests.

