## [Peer Review File · Nature Communications]

Reviewers' comments:

Reviewer #1 (Remarks to the Author):

This manuscript has reported the comparison of the effect of different carbon-based coating onto Li metal to suppress Li dendritic formation and suggested a strategic designing of a novel electrode structure with CNT coating with a gradient ZnO content. It is quite interesting that top-surface is CNT-coated with low content of ZnO which shows lithiophobic and bottom is with high content of ZnO which shows lithiophilic, resulting in suppression of further dendrite growth outside electrode and stable SEI layer onto Li foil. The authors have clearly concluded the need of lithiophobic properties of graphene or CNT to suppress the dendrite growth but the porosity is required for Li mobility into the carbon layer to deposit onto the surface of Li foil. However, following points should be addressed before accepting this manuscript for publication.

1] The authors should mention the loading materials (approximately in gram) in 100 $\mu\text{L cm}^{-2}$. Is it equal to the 0.2 mg cm^{-2} in 100 $\mu\text{L cm}^{-2}$, especially electrospun CNF materials? How the author determine the materials content inside the solution that use to create the gradient interfacial layer with different amount of solution? Why the upper layer of ZnO/CNT gradient interfacial layer still content trace amount of ZnO although the final two dripping process was only contain CNT materials?

2] The authors should mention whether all the CNT in the experiment is heat-treated at 1500 C on page 5 and state clearly that all the CNT in the following experiment, including CNT in ZnO/CNT with and without a gradient interfacial layer, is using the same.

3] Necking behaviors results from dendrite formation in early stage and SEI accumulation later on page 6, line 133. However, a number of existing papers reported decreasing high overpotentials in early stage is in stabilization step and increasing overpotentials later is due to dendrite growth, mossy Li, and so on. Please explain the difference between your work and other references.

* Liangbing Hu et al., Conformal, Nanoscale ZnO Surface Modification of Garnet-Based Solid-State Electrolyte for Lithium Metal Anodes. *Nano letters*, 2017 17 pp565

4] The authors described the different cycle stability of carbon black coating and carbon fiber coating with CNT coating on page 6, line 141. The authors should explain the reason.

5] The authors should cite the reference paper that the Zn can be alloyed with lithium, resulting in improving of lithiophilicity of CNT when adding ZnO.

*Yayuan Liu et al., Lithium-coated polymeric matrix as a minimum volume-change and dendrite-free lithium metal anode, *Nature Communications*, 2016 7 pp10992

Reviewer #2 (Remarks to the Author):

The paper describes the development of an interfacial layer for stabilization of the lithium metal interface. The strategy is based on layering lithiophilic ZnO/CNT and lithiophobic CNT layers at the surface of a lithium foil electrode. While both materials have been used in other studies to develop metallic Li anodes, this interfacial layer approach appears unique. The experimental approach, analysis and detail in this paper is good and I enjoyed reading it. The level of English was acceptable but further proofing is advised. As the focus of the paper is performance improvement, it is important that performance metrics are very clear and I propose some small additional measurements and analysis around this which would add value to the paper. I support publication with some minor additions.

Minor points below:

1) Please make clear the weight, thickness and volume of the interfacial layer and together with

the mass and volume of lithium cycled provide materials level specific and volumetric capacities for the whole electrode. See this article for guidance - www.nature.com/articles/nenergy201791

2) While cycling of lithium foil demonstrates the lack of dendrites and material stability, it is not ideal when confirming cycling stability and reversibility. It would be very interesting to see how the interfacial layer performs at a copper foil, perhaps with a realistic 3 x excess metallic lithium at the surface.

Reviewer #3 (Remarks to the Author):

This research represents a significantly important topic: manipulating a stable solid electrolyte interphase (SEI) at the lithium-metal anode for the next-generation lithium-metal batteries.

However, the presented strategy with a carbon-material based interlayer is not a novel approach, although the authors reported here a gradient interlayer concept. Therefore, it is not recommended to publish this manuscript in Nature Communications. However, the authors may consider publishing this research in another specific journal.

A few specific comments are listed below.

1. A major concern is that the carbon-based interlayer approach has been previously reported in many publications.
2. The authors may need to explain why the overpotential increases so significant after a certain time of cycling in Figure 2.
3. One curve seems missed in Figure 7e.

Dear Reviewers:

Thank you for the comments concerning our manuscript entitled “Lithiophilic/lithiophobic gradient interfacial layer for highly stable lithium metal anode”. Those comments are most valuable and helpful for revising and improving our paper, as well as providing the important guidance to our research. We have modified the manuscript accordingly, and listed the detailed corrections below point by point for each reviewer. Moreover, all revised portion has been marked in yellow in the revised manuscript.

The main corrections and the responses to the reviewers’ comments are as follows.

Reviewers' comments:**Reviewer #1:**

Overall Assessment: “This manuscript has reported the comparison of the effect of different carbon-based coating onto Li metal to suppress Li dendritic formation and suggested a strategic designing of a novel electrode structure with CNT coating with a gradient ZnO content. It is quite interesting that top-surface is CNT-coated with low content of ZnO which shows lithiophobic and bottom is with high content of ZnO which shows lithiophilic, resulting in suppression of further dendrite growth outside electrode and stable SEI layer onto Li foil. The authors have clearly concluded the need of lithiophobic properties of graphene or CNT to suppress the dendrite growth but the porosity is required for Li mobility into the carbon layer to deposit onto the surface of Li foil. However, following points should be addressed before accepting this manuscript for publication.”

Response: We appreciate the reviewer’s comments on the novelty and importance of our work. We here address the reviewer’s comments as detailed below.

Comment 1. The authors should mention the loading materials (approximately in gram) in 100 μL cm^{-2} . Is it equal to the 0.2 mg cm^{-2} in 100 $\mu\text{L cm}^{-2}$, especially electrospun fiber materials? How the authors determine the materials content inside the solution that use to create the gradient interfacial layer with different amount of solution? Why the upper layer of ZnO/CNT gradient interfacial layer still content trace amount of ZnO although the final two dripping process was only contain CNT

materials?

Response: Thank you for your careful review.

The synthesis of various upper interfacial layers can be categorized in two distinct ways. With CNT, ZnO/CNT, graphene, carbon black, or carbon fiber as the starting material, the layers are formed by suspension dripping, whose content is mainly controlled by the dripping volume. Herein, the loading material mass is 0.2 mg cm^{-2} . While the electrospun fiber serves as coating layer directly, specifically, the electrospun fiber with $20 \text{ }\mu\text{m}$ and 0.2 mg cm^{-2} was mechanically coated on the Li foil to form the electrospun fiber-coated interfacial layer, whose thickness depends on the electrospun parameters.

The cross-section of sample has been tilted slightly for the convenience of obtaining the morphology and elemental EDX spectrum of GZCNT interfacial layer, therefore, trace amount of ZnO from the middle part may be collected in the spectrum.

The related comments have been added and marked with a yellow background in **Preparation of interfacial layer coated Li foils** (page 21) and **Structural and Characterization of interfacial layers and Li foils** (page 22) in **Methods**, as follows:

“The thickness of the electrospun PI matrix is $\sim 20 \text{ }\mu\text{m}$ with 0.2 mg cm^{-2} . The electrospun fiber was dried and coated directly on the Li foil in a glove box.”

“With CNT, ZnO/CNT, graphene, carbon black, or carbon fiber as the starting material, we first dried them in vacuum at $120 \text{ }^\circ\text{C}$ for 2h, and then move them into a Ar-filled glove box, dispersed them in 1,3-dioxolane (DOL) with a concentration of 0.2 wt. % by stirring for 12h, dripped the suspensions onto the Li foils with $100 \text{ }\mu\text{L cm}^{-2}$ by a pipette and then dried by a hot plate at $80 \text{ }^\circ\text{C}$ for 1h, leading to a mass loading of 0.2 mg cm^{-2} for all the layers.”

“The cross-section of sample will tilt slightly for the convenience of obtaining the morphology and EDX spectrum.”

Comment 2. The authors should mention whether all the CNT in the experiment is heat-treated at $1500 \text{ }^\circ\text{C}$ on page 5 and state clearly that all the CNT in the following experiment, including CNT in ZnO/CNT with and without a gradient interfacial layer, is using the same.

Response: Thank you for your kind suggestion. All the CNT used in the experiment is heat-treated at $1500 \text{ }^\circ\text{C}$. This treatment, on one hand, may eliminate the disturbance from various functional groups, on the other hand, enhances the graphitization degree of CNT, which may facilitate the fast electron

transport. Notably, good electronic conductivity could reduce local current density, thus mitigating the dendritic formation. [Zuo, T. et al. Graphitized carbon fibers as multifunctional 3D current collectors for high areal capacity Li anodes. *Adv. Mater.* 29, 29 (2017).]

The related description has been added in detail in **Preparation of interfacial layer coated Li foils** (page 20) in **Methods**. We have marked the revised part in yellow in the updated manuscript:

“CNT (Cnano Tech Corp., China) used in the experiment was all heated at 1500 °C for 2h to remove the nitrogen functional groups before use and enhance the graphitization degree of CNT, which may facilitate the fast electron transport.”

Comment 3. Necking behaviors results from dendrite formation in early stage and SEI accumulation later on page 6, line 133. However, a number of existing papers reported decreasing high overpotentials in early stage is in stabilization step and increasing overpotentials later is due to dendrite growth, mossy Li, and so on. Please explain the difference between your work and other references.

* Liangbing Hu et al., Conformal, Nanoscale ZnO Surface Modification of Garnet-Based Solid-State Electrolyte for Lithium Metal Anodes. *Nano letters*, 2017 17 pp565

Response: Thank you for your valuable question. The necking behaviors, i.e. the overpotential first decreases then increases, can be explained as follows: the initial few cycles is the process of SEI formation, when the SEI is mature and becomes relatively stable after the initial process, the overpotential reduces. And then, the overpotential increases upon cycling, indicating an elevated charge-transfer resistance due to unstable Li/electrolyte interface from continuous growth of dendritic Li. The necking behaviors exhibited by our data are consistent with those reported in the literature.

The paper (Hu, L. et al., Conformal, Nanoscale ZnO Surface Modification of Garnet-Based Solid-State Electrolyte for Lithium Metal Anodes. *Nano lett.* 17, 565 (2017)) has been cited as reference 43 in our paper. The related comments have been added and marked with a yellow background on **Page 7**:

“the bare Li electrode and Li coated by electrospun fiber still exhibit remarkable necking behaviors (the overpotential first decreases, then increases) during cycling (Fig. 3c), which is a characteristic sign for stabilization of SEI in early stage and SEI accumulation from continuous growth of dendritic Li later⁴³”

Comment 4. The authors described the different cycle stability of carbon black coating and carbon fiber coating with CNT coating on page 6, line 141. The authors should explain the reason.

Response: Thank you for your kind suggestion. Carbon materials, including CNT, carbon fiber, graphene, and carbon black, are proved to be ideal candidates for Li metal anode modification. However, most of the studies focus on individual carbon material, and it is necessary to make a systematic study on their application in lithium metal anode protection to obtain the principles for designing and constructing stable Li metal anodes. Herein, it is demonstrated that the cycling stability of carbon black and carbon fiber is significantly inferior to that CNT-based interfacial layers, because both carbon fiber and carbon black have bad film forming properties, the interfacial layers formed by them are discontinuous (insets of Supplementary Fig. 7a and 8a), thus they fail to protect the Li foils from the uneven Li deposition.

The related comments have been added and marked with a yellow background on **Page 9**:

“Because both carbon fiber and carbon black have bad film forming properties, the interfacial layers formed by them are discontinuous (insets of Supplementary Fig. 7a and 8a), thus they fail to protect the Li foils from the uneven Li deposition (the carbon black layer even worsen this condition). In brief, all the interfacial layers in Fig. 4, including carbon materials and electrospun fiber, were unable to stabilize the SEI in long term, consequently resulting in increased interfacial resistance due to repeated formation of SEI.”

Comment 5. The authors should cite the reference paper that the Zn can be alloyed with lithium, resulting in improving of lithiophilicity of CNT when adding ZnO.

*Yayuan Liu et al., Lithium-coated polymeric matrix as a minimum volume-change and dendrite-free lithium metal anode, Nature Communications, 2016 7 pp10992

Response: Thank you for your useful suggestion. The as mentioned literature and another related one have been cited as reference 26 and 31, and the related changes were labeled by the yellow marker.

26. Liu, Y. Y. et al. Lithium-coated polymeric matrix as a minimum volume-change and dendrite-free lithium metal anode. *Nat. Commun.* **7**, 10992–11000 (2016).

31. Jin, C. B. et al. 3D lithium metal embedded within lithiophilic porous matrix for stable lithium metal batteries. *Nano Energy* **37**, 177–186 (2017).

Reviewer #2:

Overall Assessment: “The paper describes the development of an interfacial layer for stabilization of the lithium metal interface. The strategy is based on layering lithiophilic ZnO/CNT and lithiophobic CNT layers at the surface of a lithium foil electrode. While both materials have been used in other studies to develop metallic Li anodes, this interfacial layer approach appears unique. The experimental approach, analysis and detail in this paper is good and I enjoyed reading it. The level of English was acceptable but further proofing is advised. As the focus of the paper is performance improvement, it is important that performance metrics are very clear and I propose some small additional measurements and analysis around this which would add value to the paper. I support publication with some minor additions.”

Response: We greatly appreciate the reviewer’s encouragement and commendation on the novelty of our work. We here address the reviewer’s comments and discuss changes introduced in this revised manuscript as detailed below. Besides, the English language in our manuscript has been polished as a whole by a native speaker.

Minor points below:

Comment 1. Please make clear the weight, thickness and volume of the interfacial layer and together with the mass and volume of lithium cycled provide materials level specific and volumetric capacities for the whole electrode. See this article for guidance-www.nature.com/articles/nenergy201791

Response: Thank you for your valuable suggestion. As is reported by Stefan A. Freunberger, the capacity based on the active materials alone does not reflect true electrode performance, and it is important to present the mass and volume fractions of active material. As shown in Fig. R1 below, the thickness, volume and weight fraction of the interfacial layer was less than 0.85 %, 0.85 %, and 0.19 % respectively, which has almost no influence on the performance of the whole electrode.

Figure R1 | The thickness, volume and weight fraction of the interfacial layer and Li foil with a diameter of 16 mm.

The related comments have been added and marked with a yellow background on **Page 11**. The paper by *Freunberger* has been cited as reference 45:

“It should be noted that it is important to present the mass and volume fractions of active material for better electrode performance reflection⁴⁵. The thickness (20 μm), volume and weight (0.2 mg cm⁻²) fraction of the GZCNT layer was less than 0.85%, 0.85%, and 0.19% respectively, which has almost no influence on the performance of the whole electrode.”

45. Freunberger, S. True performance metrics in beyond-intercalation batteries. *Nat. Energy* **2**, 17091 (2017).

Comment 2. While cycling of lithium foil demonstrates the lack of dendrites and material stability, it is not ideal when confirming cycling stability and reversibility. It would be very interesting to see how the interfacial layer performs at a copper foil, perhaps with a realistic 3 x excess metallic lithium at the surface.

Response: The reviewer brought up an important point and we agree the reviewer in this regard. To better understand the role of GZCNT in suppressing dendrites, Li plating/stripping on Cu current collector with different coating layers was investigated using a half cell constructed on a Li electrode

with a capacity of 3 mA h cm^{-2} at a current density of 2 mA cm^{-2} (**Supplementary Fig. 16**).

During each discharge/charge cycle, a fixed amount of Li metal (3.0 mA h cm^{-2}) was plated onto the Cu electrode, and then stripped away. Thus, the Coulombic efficiency (CE) here quantifies the amount of Li metal recovered from the working electrode in the reverse stripping process, offering an important parameter to evaluate the cycling efficiency of batteries. As shown in Supplementary Fig. 16a, the GZCNT coated Cu electrodes exhibit greatly improved cycling stability with enhanced CEs. The CEs of GZCNT coated electrodes can be improved up to $\sim 99.5\%$ at a current density of 2.0 mA cm^{-2} with an extended cycle life of over 100 cycles. In comparison, the CEs of bare Cu electrodes start to fluctuate markedly after 30 cycles at 2.0 mA cm^{-2} , indicating the formation and sporadic activation of ‘dead Li’ and unstable anode/electrolyte interface during cycling.

The huge difference between GZCNT coated Cu electrodes and bare Cu electrode can also be reflected by the change of cell polarization (hysteresis) state with extended cycling. The voltage hysteresis of the Cu foil increases to above 50 mV only after 20 cycles, indicating continuous reaction between Li metal and electrolyte along with increasing internal resistance caused by the thickened SEI films. While the voltage hysteresis of the GZCNT coated Cu foil cell is maintained less than 15 mV after 100 cycles (Supplementary Fig. 16b-d), and this demonstrates its promising prospect in high energy batteries. By contrast, the ZCNT coated Cu layer, which is lithiophilic, displays lower cell polarization, while Li can be easily deposited on the surface of CNT with uniform deposition of ZnO, and then the interspace in the layer is slowly stuffed by Li deposits as the cycles continue (Supplementary Fig. 16c). Mossy Li tends to form on the stuffed ZCNT interfacial layer, leading to the failure of the cell (Supplementary Fig. 16f). The CE enhancement of ZCNT coated Cu electrode thereby was very limited compared with plain Cu, which can also be confirmed by the morphology of the deposited Li metal on these current collectors after different cycles.

According to the top view SEM images, after long-term cycling, many long Li filaments can be observed for the plain Cu current collector, and the evolution of this filament aggregated structure also leads to the uneven distribution of the electrical field, further accelerating inhomogeneous Li deposition, which may pierce the separator and cause a safety hazard. In comparison, for a GZCNT coated Cu layer, the morphology of CNT is well retained, and no dendrite was visible. To further reveal the Li deposition process of GZCNT coated Cu layer, we show SEM images of cross-section

view morphology of pristine GZCNT coated Cu foil after stripping and plating back 3 mAh cm^{-2} , and after 500 cycles at 2 mA cm^{-2} with various magnifications. The GZCNT interfacial layer tightly anchored on the surface of Li deposits on Cu current collector (Supplementary Fig. 16g and h). In contrary to the porous structure on the top of GZCNT, the interfacial layer at the bottom is very dense and homogenous (Supplementary Fig. 16i). The forming of this compact interface is due to stable and uniform SEI from the lithiophilic layer, which effectively ensured the uniform Li plating/stripping.

In brief, the gradient GZCNT layer was further demonstrated to be successful in regulating the deposition of Li on Cu current collector. The top lithiophobic part of GZCNT layer kept a porous morphology to facilitate the Li diffusion and hinder the dendrite forming, while the bottom lithiophilic layer anchored can effectively ensure an evenly Li plating by building a stable SEI, resulting in the fact that neither dendrites shoot out of the electrode nor the corrosion layer between CNT and Cu foil are formed.

Supplementary Figure 16 | The characterization of Cu|Li asymmetrical cells with pristine, ZCNT and GZCNT coated Cu electrodes. (a) Comparison of cycling stability of pristine Cu electrodes (red) and ZCNT (blue) and GZCNT coated Cu electrodes (dark yellow) at current density of 2.0 mA cm⁻² with a capacity of 3 mA h cm⁻². Voltage profiles of the Li plating/stripping process

on (b) bare Cu electrodes, (c) ZCNT and (d) GZCNT coated Cu electrodes with a capacity of 3 mA h cm⁻² at a current density of 2 mA cm⁻². SEM images of top view morphology of bare Cu electrode (e), ZCNT (f) and GZCNT coated Cu electrodes (g) after (e) 30, (f) 40, and (g) 110 cycles at 2 mA cm⁻² with 3 mAh cm⁻² stripping/plating capacity. The insets show the digital photos of the corresponding Cu current collectors with interfacial layers after test. The cross-section view morphology of GZCNT coated Cu electrode at low (h) and high (i) magnifications, showing a smooth surface with well-preserved thin layer of SEI.

The related comments have been added and marked with a yellow background on **Page 15, Supplementary Figure 16 and related note:**

“It is demonstrated that the gradient strategy also excelled at a copper current collector, which achieves an enhanced cycling stability, low voltage hysteresis, and a higher average Coulombic efficiency of 99.5% within 100 cycles, with a capacity of 3 mA h cm⁻² at a current density of 2 mA cm⁻² (Supplementary Fig. 16).”

Reviewer #3:

Overall Assessment: “This research represents a significantly important topic: manipulating a stable solid electrolyte interphase (SEI) at the lithium-metal anode for the next-generation lithium-metal batteries. However, the presented strategy with a carbon-material based interlayer is not a novel approach, although the authors reported here a gradient interlayer concept. Therefore, it is not recommended to publish this manuscript in Nature Communications. However, the authors may consider publishing this research in another specific journal.”

Response: Thank you very much for your comments.

We for the first time proposed a lithiophilic/lithiophobic gradient strategy, of which the bottom lithiophilic layer tightly anchoring onto the Li foil, facilitates the formation of a stable solid electrolyte interphase, eliminates the mossy Li corrosion layer between them, and enhances uniform Li deposition even at a high current density with a high Li capacity and long cycle life. While the top lithiophobic layer is robust enough to effectively hinder the dendrite penetrating and highly porous to facilitate the Li diffusion, and then Li dendrites hardly reach the upper surface of the composite electrode. To elucidate the lithiophilic/lithiophobic gradient strategy more clearly, we have added a schematic diagram for Li deposition of GZCNT interfacial layer coated Li foils as shown in Figure 1.

Figure 1 | Schematic diagram for Li deposition of GZCNT interfacial layer coated Li foils. The bottom lithiophilic part of GZCNT layer anchored onto the Li foil and effectively ensured an evenly Li plating by regulating deposition, while the top lithiophobic part kept a porous morphology to facilitate the Li diffusion and hinder the dendrite forming, which results in dendrite-free Li metal

anode.

We demonstrate that this lithiophilic/lithiophobic gradient strategy is unique by comparing it to various Li metal anode modification strategies using carbon materials, which is summarized in **Table R1**. Most strategies focus on either regulating Li deposition or SEI stabilization. For the gradient strategy in our work, the bottom lithiophilic layer facilitates a stable SEI, while the top lithiophobic layer is strong enough to suppress dendrite growth, achieving a synergistic effect by integrating multiple advances. When applying this strategy to CNT layer coated Li foil, no visible dendrites were observed even under a high current density of 10 mA cm^{-2} , and then an ultra-long term stable Li stripping/plating (more than 200 cycles) was obtained, which was superior to most results reported in literature.

It is also demonstrated that this lithiophilic/lithiophobic gradient strategy is not only applicable to carbon materials like CNT for Li anode modification, but also fits well in other materials. Here we demonstrate its successful application on electrospun fibers (**Supplementary Fig. 17, 18 and 19**) as is detailed presented in the response to the **Comment 1 of Reviewer 3#** listed below, which achieves an enhanced cycling stability and lower voltage hysteresis.

Table R1 Different Li metal anode modification strategies using various carbons.

Strategies	Merits	Demerits	Carbon materials involved	Reference
High SSA hosts	Decreasing local current density;	Inducing more	3D graphene foam	1
	Accommodating volume changes	irreversible reactions	rGO	2
As lithium hosts	Regulating Li deposition	Insufficient structure stability	3D CNF network enriched with oxygen containing groups	3
			Semi-conducting SiC-coated carbon-fiber papers	4
			CNTs surface covered with Al_2O_3	5
Decorated by other "lithiophilic" materials	Ensuring uniform	the Larger Li volume	Decorating carbon with Si, Sn, Au, Al,	6-8

		deposition	changes of “lithiophilic” materials	Mg, Zn or Ag	
As interfacial layers	SEI stabilization	Protecting Li metal directly	Impeding ion diffusion	A monolayer of amorphous hollow carbon nanospheres	9
				Ultrathin graphene	10
	Ionic concentration adjustment	Decreasing ion concentration polarization	Large volume occupation and increasing impedance	Graphene nanosheets doped with both nitrogen and sulfur coated on the separator	11
	Lithiophilic/lithiophobic Gradient interfacial layer	Bottom lithiophilic layer facilitates a stable SEI, while the top lithiophobic layer is strong enough to suppress dendrite growth		Gradient ZnO-coated CNT	This work

1. Cheng, X. B. et al. Dual-phase lithium metal anode containing a polysulfide-induced solid electrolyte interphase and nanostructured graphene framework for lithium–sulfur batteries. *ACS Nano*, **9**, 6373-6382 (2015).
2. Lin, D. et al. Layered reduced graphene oxide with nanoscale interlayer gaps as a stable host for lithium metal anodes. *Nat. Nanotechnol.* **11**, 626 (2016).
3. Zhang, A., Fang, X., Shen, C., Liu, Y. & Zhou, C. A carbon nanofiber network for stable lithium metal anodes with high Coulombic efficiency and long cycle life. *Nano Research* **9**, 3428-3436 (2016).
4. Ji, X. et al. Spatially heterogeneous carbon-fiber papers as surface dendrite-free current collectors for lithium deposition. *Nano Today* **7**, 10-20 (2012).
5. Zhang, Y. et al. A carbon-based 3D current collector with surface protection for Li metal anode. *Nano Res.* **10**, 1356-1365 (2017).
6. Yan, K. et al. Selective deposition and stable encapsulation of lithium through heterogeneous seeded growth. *Nat. Energy* **1**, 16010 (2016).
7. Jin, C. et al. 3D lithium metal embedded within lithiophilic porous matrix for stable lithium metal batteries. *Nano Energy* **37**, 177-186 (2017).
8. Zhang, Y. et al. High-capacity, low-tortuosity, and channel-guided lithium metal anode. *PNAS* **114**, 3584-3589 (2017).
9. Zheng, G. et al. Interconnected hollow carbon nanospheres for stable lithium metal anodes. *Nat. Nanotechnol.* **9**, 618 (2014).

10. Yan, K. et al. Ultrathin two-dimensional atomic crystals as stable interfacial layer for improvement of lithium metal anode. *Nano Lett.* **14**, 6016-6022 (2014).
11. Shin, W., Kannan, A. & Kim, D. Effective Suppression of Dendritic Lithium Growth Using an Ultrathin Coating of Nitrogen and Sulfur Co-doped Graphene Nanosheets on Polymer Separator for Lithium Metal Batteries. *ACS Appl. Mater. Interfaces* **7**, 23700-23707 (2015).

More remarkably, this gradient strategy can also be extended to Cu current collector for stable Li plating/stripping as is demonstrated in the response to **Comment 2 of Reviewer 2**. The homogeneous and stable SEI from this strategy helps to realize a high Coulombic efficiency (>99.5%) with a dendrite-free Li depositing morphology (**Supplementary Fig. 16**).

In conclusion, the gradient strategy has been proved to be unique and very powerful to suppress Li dendrites by multiple ways, and the universality will make this strategy promising for practical Li metal usage.

A few specific comments are listed below.

Comment 1. A major concern is that the carbon-based interlayer approach has been previously reported in many publications.

Response: Thank you very much for your valuable comments. In this work, the key is that we demonstrate a promising lithiophilic/lithiophobic gradient interfacial layer strategy, in which the bottom lithiophilic sublayer, tightly anchoring the whole layer onto the Li foil, facilitates the formation of a stable solid electrolyte interphase and contributes to a uniform Li plating. When combined with the top lithiophobic sublayer, this gradient interfacial layer can effectively suppress dendrite growth.

Compared with other strategies of the Li metal anode modification using carbon materials reported (**Table R1**), this gradient strategy was proposed unprecedentedly, achieving the integration of SEI stabilization and regulating the Li deposition. Notably, the gradient strategy is not only applicable to carbon materials like CNT, but also fits well in other materials for Li anode modification.

In light of this, the strategy was typically explored in a gradient ZnO/electrospun fiber (GZF) interfacial layer for Li-metal anodes. Herein, the Galvanostatic cycles of Li|Li symmetrical cells were conducted to probe the long-term cycling stability of GZF layer-coated Li electrode. Excitingly,

the GZF-Li|GZF-Li cell displays an excellent cycling stability as evidenced by a negligible voltage fluctuation and a much lowered overpotential (~80 mV in the initial cycles, and further stabilizes at 45 mV for more than 500 cycles (~1000 h)) (**Supplementary Fig. 17c**). During the cycling process, a quick decrease of the hysteresis indicates a fast SEI formation, after cycling for over 1000 h, no sign of voltage oscillations is observed. Even under higher current densities of 5 mA cm⁻², the GZF-Li|GZF-Li cell, with a corresponding overpotential of 100 mV (**Supplementary Fig. 17d**), can still run for 135 h. These results suggest that a stable and homogeneous SEI forms between the GZF interfacial layer (**Supplementary Fig. 17g-h**) and Li, which ensures enhanced charge-transfer kinetics. In brief, this gradient strategy was further validated efficient in electrospun fiber (**Supplementary Fig. 17-19**).

Even more exciting, this gradient strategy can also be extended to Cu current collector for stable Li plating/stripping. The homogeneous and stable SEI from this strategy helps to realize a high Coulombic efficiency (>99.5%) with a dendrite-free Li depositing morphology (**Supplementary Fig. 16**).

Therefore, the gradient strategy has been proved to be unique and very powerful to suppress Li dendrites by multiple ways, and its universality will make this strategy promising for practical Li metal usage.

The related comments have been added and marked with a yellow background on **Page 15** and **Supplementary Figure 17-19**:

“The strategy may not be limited to various carbon materials like CNT, it may be extended to other materials for Li metal modifications. Here the strategy was typically applied to the gradient ZnO/electrospun fiber (GZF) interfacial layer, and proved equally effective in guiding the deposition of Li metal. As shown in Supplementary Fig. 17, the bottom lithiophilic layer anchored tightly onto the Li foil, effectively facilitating a stable SEI formation and regulating the Li deposition. Together with the top lithiophobic CNT, this GZF interfacial layer can effectively suppress dendrite growth even under a high current density of 5 mA cm⁻², and ensure an ultra-long term stable Li stripping/plating.”

Supplementary Figure 17 | Symmetrical cell testing of bare Li electrode and Li foils with

various fiber-based interfacial layers. (a) Schematics of the cell design. **(b)** Electrochemical impedance spectra of various cells. Frequency: 0.1 Hz–1 MHz, perturbation amplitude 5 mV, measured at 0% state of charge (SOC). **c-d**, Comparison of the cycling stability of a symmetric cell assembled by blank Li foils (red), and Li foils with interfacial layers of electrospun fiber (blue), ZF (green), and GZF (dark yellow), respectively, at charge/discharge current densities of **(c)** 1 and **(d)** 5 mA cm⁻² with a stripping/plating capacity of both 1 mAh cm⁻². SEM images of top view morphology of electrospun fiber **(e)**, ZF **(f)** and GZF coated Li **(g)** after **(e)** 220, **(f)** 50, and **(g)** 500 cycles at 1 mA cm⁻² with 1 mAh cm⁻² stripping/plating capacity. The insets show the digital photos of the corresponding Li foils with interfacial layers after test. **(h)** Cross-section view morphology of GZF coated Li foil at high magnifications, showing a smooth surface with well-preserved thin layer of SEI.

Comment 2. The authors may need to explain why the overpotential increases so significant after a certain time of cycling in Figure 2.

Response: Thank you for your question. During Li plating/stripping, the overpotential increases significantly (new Figure 3), especially for pristine Li foil, which could possibly be caused by the unstable Li/electrolyte interface and electrical disconnection because of repeated growth/corrosion of dendritic Li. After some modification by electrospun fiber, graphene, the cycle stability was improved but very limited, and the overpotential finally increases significantly since they fail to stabilize the SEI in long term. For lithiophilic ZnO/CNT interfacial layers coated Li foils, Li can be deposited evenly, however, they prefer to deposit on the upper surface of the interfacial layer and form lithium dendrites continuously, consequently resulting increased interfacial resistance due to repeated formation of SEI.

The related comments have been added and marked with a yellow background on **Page 9**:

“Because both carbon fiber and carbon black have bad film forming properties, the interfacial layers formed by them are discontinuous (insets of Supplementary Fig. 7a and 8a), thus they fail to protect the Li foils from the uneven Li deposition (the carbon black layer even worsen this condition). In brief, all the interfacial layers in Fig 4, including carbon materials and electrospun fiber, were unable to stabilize the SEI in long term, consequently resulting in increased interfacial resistance due to repeated formation of SEI.”

Comment 3. One curve seems missed in Figure 7e.

Response: Thank you for your gentle comment. We have integrated Figure 7d and e together to be new Figure 8d for easy reading:

Figure 8 | **Electrochemical performance of 10 cm² pouch cells assembled by Li foils with GZCNT interfacial layers.** (a) Digital photos of two 40×25 mm² GZCNT coated Li foils and (b) the pouch cell assembled by them. (c) Comparison of the cycling stability of symmetric pouch cells assembled by blank Li foils (red), and Li foils with GZCNT (blue), respectively, at charge/discharge current density of 1 mA cm⁻² (10 mA in total) with a stripping/plating capacity of 1 mAh cm⁻² (10 mAh in total) for as long as 210 cycles. The insets are digital photos of Li foil (left) and GZCNT coated Li foils (right) in pouch cells after 100 and 200 cycles, respectively. (d) Electrochemical impedance spectra of pristine pouch cells assembled by bare Li foils and GZCNT coated Li foils, and after 100 and 200 cycles. Frequency: 0.1 Hz–1 MHz.

REVIEWERS' COMMENTS:

Reviewer #1 (Remarks to the Author):

The authors had revised this manuscript according to the reviewers' comments. The reviewer generally understands the answers in the response letter and the revised manuscript. Therefore, I think that this manuscript is sufficient to be published in Nature Communications.

Reviewer #2 (Remarks to the Author):

The authors have answered all points to a high standard and I support publication.

Dear Reviewers:

We greatly appreciate your encouragement and commendation on the novelty and importance of our work entitled “Lithiophilic/lithiophobic gradient interfacial layer for highly stable lithium metal anode”. Those comments are valuable and helpful for driving us to go forward!

REVIEWERS' COMMENTS:

Reviewer #1 (Remarks to the Author):

The authors had revised this manuscript according to the reviewers' comments. The reviewer generally understands the answers in the response letter and the revised manuscript. Therefore, I think that this manuscript is sufficient to be published in Nature Communications.

Response: Thank you very much for your positive comments to our work.

Reviewer #2 (Remarks to the Author):

The authors have answered all points to a high standard and I support publication.

Response: Thank you very much for your great encouragement to our work.